# Impacts of COVID-19 Travel Restriction Policies on the Traffic Quality of the National and Provincial Trunk Highway Network: A Case Study of Shaanxi Province

**DOI:** 10.3390/ijerph19159387

**Published:** 2022-07-31

**Authors:** Yongji Ma, Jinliang Xu, Chao Gao, Xiaohui Tong

**Affiliations:** 1School of Highway, Chang’an University, Xi’an 710064, China; gaochao@chd.edu.cn; 2China Communications Construction First Highway Consultants Co., Ltd., Xi’an 710061, China; tongxiaohui@ccccltd.cn

**Keywords:** COVID-19, travel restriction policies, the national and provincial trunk highway network, service level, safety level, operational orderliness, traffic quality

## Abstract

According to recent research, the COVID-19 pandemic has impacted road traffic quality. This study aims to analyze the impacts of COVID-19 travel restriction policies on the traffic quality of the national and provincial trunk highway network (NPTHN) in Shaanxi Province. We collected the traffic data of the NPTHN for three consecutive years (from 2019 to 2021), before and after the COVID-19 outbreak, including weekly average daily traffic, weekly traffic interruption times, weekly traffic control time, weekly traffic accidents, weekly traffic injuries, and weekly traffic deaths. Using descriptive statistics and dynamic analysis methods, we studied the safety and service levels of the NPTHN. We set up an assessment model of the NPTHN operational orderliness through dissipative structure theory and entropy theory to study the operational orderliness of the NPTHN. Results show that in 2020, the service level, safety level, and operational orderliness of the NPTHN dropped to the lowest levels. The pandemic was gradually brought under control, and the travel restriction policies were gradually reduced and lifted. The adverse impacts on the operational orderliness of the NPTHN decreased, but the operational orderliness did not yet recover to the pre-pandemic level. Meanwhile, the service and safety levels of the NPTHN did not recover. Taken together, the COVID-19 travel restriction policies had adverse impacts on the traffic quality of the NPTHN in Shaanxi Province.

## 1. Introduction

The COVID-19 outbreak has brought enormous challenges to the social development of countries globally [1,2,3,4]. China has accumulated more than 3.8 million confirmed cases and more than 19,000 deaths since the beginning of the COVID-19 outbreak [5]. Faced with the threat of the COVID-19, China and other countries adopted various pandemic prevention and control measures [6,7,8,9,10] to protect the residents’ health and lives. Some of the measures reduced people’s travel frequency and people’s frequency of road-use in a certain period [11,12,13,14]. Under this background, the traffic operation characteristics of the national and provincial trunk highway network (NPTHN) would inevitably change. Moreover, researchers carried out extensive studies to explore the impacts of COVID-19 and its prevention and control measures on road traffic.

Gu et al. demonstrated that the origin-destination (OD) flows associated with passenger cars on intercity highways in China decreased significantly during COVID-19. With the effective implementation of the pandemic prevention control policy and the orderly promotion of the recovery to work and production, the volumes of intercity highway OD flows gradually returned to the pre-pandemic levels [15]. Li et al. proposed an equation to quantify the change in traffic volume in the whole province. They also analyzed the change in intercity bus flow and the use of intercity buses and demonstrated the impact of COVID-19 and its traffic restriction measures on provincial traffic [16]. Zhang et al. analyzed the changes in the local travel behavior of various population groups before and during the COVID-19 pandemic in Hong Kong [17].

In addition, Adanu et al. studied the impact of COVID-19 on road traffic accidents and their severity under the background of traffic restriction policies by establishing four crash-severity models. They found that although the road traffic volume and vehicle mileage during the lockdown decreased significantly, the total number of crashes and major injury crashes increased compared with the period prior to the lockdown [18]. Mokhtarimousavi et al. studied the relationship between the lockdown during the COVID-19 pandemic and the severity of injuries sustained by drivers involved in run-off-road crashes [19]. Li et al. studied the effect of COVID-19-related travel restriction policies on road traffic accident patterns and focused on the safety of cyclists [20]. Qureshi et al. demonstrated that the mandated societal lockdown policies led to a reduction in road traffic accidents resulting in non-serious or no injuries but not those resulting in serious or fatal injuries [21]. Nomura et al. found that the number of road traffic deaths during the COVID-19 pandemic in Japan declined slightly over several weeks compared with the average year, but not significantly [22]. Shimizu et al. assessed traffic accident rates from April 2020 to December 2021 [23]. Adanu et al. investigated the differences and similarities in the effects of the factors associated with road traffic crash injury severity before, during, and after the shelter-in-place order [24]. Tucker et al. found that the percentage of cars engaged in extreme speeding and those traveling below the speed limit increased during the COVID-19 stay-at-home period [25]. Katrakazas et al. made a quantitative assessment of the effect of COVID19 on driving behavior during the lockdown [26]. Vanlaar et al. compared self-reported risky driving behaviors during the pandemic in Canada and the U.S. to determine the differences between these two countries [27].

The impacts of the COVID-19 pandemic and its traffic restriction policies on road traffic quality cannot be ignored. Existing studies mainly focused on the topic of traffic safety in the COVID-19 context, and few of them involved specific traffic parameters and traffic behavior analysis. Research on the provincial road traffic quality from the perspective of multiple indicators and multiple parameters is limited, which is exactly the main problem that this study aims to solve. In addition, road traffic is an example of very complex system engineering, and many methods and theories exist to study road traffic. Among them, as theories to study the developmental state of a material system, entropy theory and dissipative structure theory have been applied by researchers in the field of traffic engineering. Chai et al. used information entropy theory to establish a composite element evaluation model for urban mountain road safety [28]. Sun et al. established an urban traffic congestion prediction model through the hidden Markov model combined with the dissipative structure theory and entropy theory [29]. Deng et al. applied the information entropy theory to the study of road traffic safety evaluation in Jiangsu Province and obtained reasonable evaluation results [30]. Hence, we chose entropy theory and dissipative structure theory as part of the research methods in this study.

From the traffic engineering perspective, road traffic quality includes three main aspects: the road service level, safety level, and operational orderliness, which are the three main perspectives for evaluating road traffic quality. Therefore, taking the service level, safety level, and operational orderliness as the points of penetration, this study attempts to analyze the traffic quality of the NPTHN in the COVID-19 context. Using statistical analysis and modeling, this study discussed the impacts of the COVID-19 travel restriction policies on the traffic quality of the NPTHN. Through quantitative analysis, the adverse impacts of COVID-19 travel restriction policies on the traffic quality of the NPTHN were identified. The results show that in traffic quality research, we should not only pay attention to the micro level of traffic factors but also should not ignore the impact of macro policies on traffic. The conclusion of this study could provide some theoretical references for the decision-making of traffic management departments in improving road traffic quality.

## 2. Materials and Methods

In 2021, the NPTHN of Shaanxi Province included 25 national trunk highways and 272 provincial trunk highways. Figure 1 depicts an overview of its geographical location.

We collected the traffic operation data of the NPTHN of Shaanxi Province for three consecutive years (from 2019 to 2021) before and after the COVID-19 outbreak, including weekly average daily traffic (WADT), weekly traffic interruption times (WTIT), weekly traffic control time (WTCT), weekly traffic accidents (WTA), weekly traffic injuries (WTI), and weekly traffic deaths (WTD). Among them, WTIT refers to the number of traffic interruptions caused by traffic accidents in the NPTHN of Shaanxi Province in a week. The higher the value, the more frequent the NPTHN interruptions in that week, and vice versa. WTCT indicates the traffic control time caused by traffic accidents in the NPTHN of Shaanxi Province in a certain week. When a road accident occurs, to ensure accident rescue and orderly traffic, the traffic management department will temporarily take measures to control the accident road section, restrict the passage of certain vehicle types, and take one-way alternate passage measures, among others. The total duration of traffic control in a week is WTCT.

The NPTHN’s operational orderliness examined in this study refers to the degree to which the traffic flow can operate continuously, stably, and smoothly under certain traffic conditions. If the traffic flow can operate continuously, stably, and smoothly, the NPTHN’s operational orderliness is high and the traffic conditions are good. Otherwise, the operational orderliness is low, and the traffic conditions are poor.

Taking WADT, WTIT, and WTCT as the NPTHN service level assessment parameters, and WTA, WTI, and WTD as the NPTHN safety level assessment parameters, the impacts of the COVID-19 travel restriction policies on the service and safety levels of the NPTHN were studied through the methods of dynamic analysis and descriptive statistics.

Taking WADT as the core parameter and WTIT as the system entropy weight parameter, an assessment model for the operational orderliness of the NPTHN was established through dissipative structure theory and entropy theory. Moreover, the entropy changes of the NPTHN system before and after the COVID-19 outbreak were analyzed.

### 2.1. Materials

The traffic operation data of the NPTHN were collected from 228 continuous observation sites in Shaanxi Province. These 228 continuous observation sites are mainly distributed in some key road sections of NPTHN in Shaanxi Province, such as intersections with large traffic volumes and important towns and bridges. The automatic traffic flow observation instrument was mainly used to count the traffic volumes of the sections; it can record the two-way traffic flow of each road section of the NPTHN all day. Its accuracy can reach more than 95%. The traffic volume data obtained were finally published on the official website of the Department of Transport of Shaanxi Province. Other traffic data investigated in this study were counted by local traffic management departments and finally reported to the provincial traffic department for summary. These data were released together with traffic volume data. All traffic interruptions and traffic controls were caused by traffic accidents rather than other conditions. These collected data covered all sections of the NPTHN and could reflect the overall traffic operation quality of the NPTHN in Shaanxi Province within a certain period of time.

Affected by the COVID-19 travel restriction policies and other uncontrollable factors, we did not obtain the complete traffic operation data for the whole year. For 2019, 2020, and 2021, traffic data of 45, 34, and 39 weeks were collected, respectively. We numbered the data of each week in each year in chronological order and finally screened out the data of 78 weeks (26 weeks each year) with overlapping time points. We obtained the time series analysis dataset of the NPTHN traffic operation quality, as shown in Table 1 and Table 2. These data overlap at time points, and the number of samples within the year is consistent, which can provide more reliable data support for analyzing the impacts of COVID-19 travel restriction policies on the traffic quality of the NPTHN. In addition, according to the records of the traffic management departments in Shaanxi Province, the weather conditions in the period to which these data belonged were similar, consisting of mainly sunny and cloudy days. Moreover, there were no major holidays, so the negative impacts of adverse weather and major holidays on the traffic quality of the NPTHN were excluded, which could ensure the typicality and credibility of the traffic data.

### 2.2. Methods

The NPTHN is a dynamically evolving open system that covers drivers, vehicles, roads, environment, traffic management, and other elements. During the operation of the NPTHN, vehicle driving is affected by factors such as road geometrics, weather conditions, and traffic management. Drivers continuously obtain information from the external environment to control the operating state of vehicles. Vehicle behaviors, such as lane-changing, car-following, and overtaking, are highly random, leading to the dynamic evolution of road traffic. Based on these traffic characteristics of the NPTHN, we chose dissipative structure theory and entropy theory to study the impacts of COVID-19 travel restriction policies on the traffic quality of the NPTHN in Shaanxi Province.

#### 2.2.1. Entropy Theory and Dissipative Structure Theory

As a state function, entropy can characterize the collective nature of several microscopic factors that constitute a system from a macroscopic level, reflecting the degree of disorder in the system. The higher the entropy of the system, the higher the degree of the disorder [31]. For an open system, the total entropy change, dS, usually consists of two parts:The entropy increase (diS) caused by irreversible processes inside the system;The entropy change (deS) generated when the system exchanges matter and energy with the external environment.

The total change in entropy of the system is the sum of the two parts [32]:dS = d_i_S + d_e_S(1)

In physics, a system can form a dissipative structure if the following four conditions are met simultaneously [33]:The system is open;The system is far away from the equilibrium state;There are nonlinear interactions in the system;There are fluctuations in the system.

The NPTHN has four basic conditions for becoming a dissipative structure, so the relevant theories are applicable. According to the dissipative structure theory, the evolution of system orderliness can be measured by the system entropy change [34]:When dS > 0, the total entropy of the system increases, and the system moves away from the equilibrium state and develops disorder;When dS < 0, the total entropy of the system decreases, and the system approaches the equilibrium state and develops in an orderly direction;When dS = 0, the total entropy of the system remains unchanged, and the system can maintain a dynamic equilibrium state.

The thermodynamic-like method and statistical-physics-like method are the two main methods for calculating a system’s entropy [34]. The former uses the ratio of certain extensive and intensity quantities in the system to calculate the entropy value, and the latter calculates the entropy value through the information theory. The NPTHN system is a dissipative structure and an information system that can be analyzed through probability theory and mathematical statistics. Thus, we finally chose the method of information theory to calculate the system entropy [35]. According to Shannon’s information theory, if the output of an information source is finite or contains only listable infinite discrete random variables, then it is discrete. Moreover, the NPTHN system can be studied as a discrete information source. For a discrete information source X, let xi be the possible output signal of X (*x_i_* ∈ X, *i* = 1, 2, 3, …, *n*), and its probability is *p*(*x_i_*). Then, the system entropy value H(x) can be calculated according to the following formula [36]:(2)$$H(x)=-C\sum_{i=1}^{n}p(x_{i})\;\log_{10}\;p\left(x_{i}\right)=\sum_{i=1}^{n}p\left(x_{i}\right)I\left(x_{i}\right),$$
where *C* is a constant, *I*(*x**_i_*) is the self-information of *x**_i_*, and the *I*(*x**_i_*) can be calculated through the following formula:(3)$$I\left(x_{i}\right)=-C\;\log_{10}\;p\left(x_{i}\right).$$

The unit of *I*(*x**_i_*) is the hartley.

#### 2.2.2. Assessment Model of the NPTHN’s Operational Orderliness

The first task of an operational orderliness assessment is to determine the order parameter for the model, because this influences the system evolution, impacts other variables’ changes, and determines the orderliness of the system to some extent [35]. With regard to the NPTHN system, traffic volume is the most critical characteristic parameter, which directly determines the operational orderliness of the NPTHN, affects other traffic parameters, such as traffic density and traffic flow speed, and reflects the regional economy and social developments to a certain degree. In this case, the order parameter ought to reflect the characteristics of traffic volume when assessing the NPTHN’s operational orderliness.

Moreover, operational orderliness is related to various factors, such as traffic volume, traffic accidents, traffic interruptions, and weather conditions. The mileage of the NPTHN and the ownership of private cars in Shaanxi Province have been growing continuously, factors which have non-negligible impacts on the traffic volume of the NPTHN. The absolute value of traffic volume is not enough to reflect the NPTHN operational orderliness level.

Taking all the above into consideration, we finally used the NPTHN patency degree *λ* as the order parameter. We defined *λ* as the ratio of daily average traffic volume within a certain period of a year to the annual average daily traffic of the same year. For the NPTHN system at week *i* of a year, the patency degree *λ* can be calculated according to the following formula:(4)$$\lambda_{i}=\frac{W_{i}ADT}{AADT},$$
where *i* is the weekly sequence number, and *W_i_ADT* is the WADT of week *i*. If *λ_i_* = 1, the level of traffic volume in week *i* is consistent with the annual average level. If *λ_i_* < 1, the level of traffic volume in week *i* is lower than the annual average level. If *λ_i_* > 1, the level of traffic volume in week *i* is higher than the annual average level. In this case, we chose *λ_i_* = 1 as the threshold of the NPTHN patency degree.

We introduced the NPTHN patency degree deviation Δ*λ_i_* to calculate the entropy value of the NPTHN system, where Δ*λ_i_* is defined as follows:(5)$$\Delta \lambda_{i}=max\left[\lambda_{i}-1,\;0\right].$$

Evidently, the patency degree deviation is 0 when *λ_i_* < 1. Let *R_i_* be the NPTHN patency degree deviation sequence, *P_i_* be the mapping sequence of *R_i_*, and *P_i_* and *R_i_* are, respectively, defined as follows:(6)$$R_{i}=\left\{\Delta \lambda_{i}\;|\;i=1,\;2,\;3,\ldots ,\;n\right\},$$
(7)$$P_{i}=\left\{\Delta \lambda_{i}/\overset{n}{\underset{i=1}{\sum }}\Delta \lambda_{i}|\;i=1,\;2,\;3,\ldots ,\;n\right\},$$
where 0 ≤ *P_i_* ≤ 1, $\sum_{i=1}^{n}P_{i}=1$, and *n* is the number of weeks in the studied year, *n* = 26.

For the NPTHN system, the entropy *S**_i_*(*t*) in week *i* can be calculated through the following formula:(8)$$S_{i}(t)=-C\times K_{i}\times P_{i}\times \log_{10}\;P_{i},$$
where *C* is a constant and *K**_i_* is the system entropy weight in week *i*. As traffic interruption represents the failure of traffic function and influences the traffic quality of the adjacent roads, it impacts the NPTHN’s operational orderliness most. Based on this, we finally chose the WTIT of the NPTHN as the key factor to calculate the system entropy weight of each week. The calculation formula of *K**_i_* is as follows:(9)$$K_{i}=\frac{W_{i}TIT}{\sum_{i=1}^{n}W_{i}TIT},$$
where *W**_i_TIT* is the WTIT in week *i* and $\sum_{i=1}^{n}K_{i}=1$. Based on this, the annual system entropy *S*(*t*) of the NPTHN in a certain year is as follows:(10)$$S(t)=-C\overset{n}{\underset{i=1}{\sum }}(K_{i}\times P_{i}\times \log_{10}\;P_{i}).$$

The entropy is essentially a relative quantity rather than an absolute quantity, and the function of constant *C* is mainly to adjust the order of magnitude of the system entropy. Therefore, *C* = 100 was taken in this research to facilitate the display of the entropy calculation results.

## 3. Results

The COVID-19 pandemic caught the world by surprise at the beginning of 2020. Faced with the threat of the pandemic, all provinces in China took positive measures to strictly prevent and control the spread of COVID-19. In 2020, the Shaanxi provincial government successively activated the first- and third-level public health emergency responses. During this time, various traffic control measures were adopted on the NPTHN in the whole province, which greatly impacted the traffic quality of the NPTHN.

For example, only the anti-pandemic-related vehicles could drive on the NPTHN in the province during the implementation of the first-level public health emergency response. The movement of private cars, commercial passenger vehicles, and freight vehicles was strictly restricted and prohibited. Some people in urgent need of medical assistance had to travel by public ambulance, and their traffic routes were strictly controlled. During the implementation of the third-level public health emergency response, people’s travel restrictions were eased, but they were still not encouraged. Reducing gatherings and avoiding unnecessary travel were still the main pandemic prevention initiatives. Meanwhile, transportation in some high-risk areas was still strictly controlled. Vehicle travel in the medium- and high-risk regions was strictly controlled and restricted during the following period of routine pandemic prevention and control, but traffic in low-risk areas was not strictly controlled. Vehicles in high-risk areas were required not to drive out of the area, and other vehicles were also prohibited from entering medium- and high-risk areas.

As a result, traffic in some areas was blocked. Under such circumstances, the COVID-19 travel restriction policies impacted the traffic quality of the NPTHN to a certain extent. Based on the collected data, we analyzed the traffic quality of the NPTHN in Shaanxi Province from the perspectives of the service level, safety level, and operational orderliness.

### 3.1. Analysis of the NPTHN Service Level

Figure 2, Figure 3 and Figure 4 show the boxplots of the NPTHN service level assessment parameters (WADT, WTIT, and WTCT). Figure 5, Figure 6 and Figure 7 depict the dot-line plots of the fixed base growth rate. Table 3 shows the descriptive statistical results of WADT, WTIT, and WTCT, and Table 4 presents the analysis results of the fixed base growth rate.

The analysis results in Figure 2 and Table 3 show that the box width and the 1.5IQR range in 2019 were narrower than those in 2020. This result indicates that before the COVID-19 outbreak, the distribution of traffic volume in the NPTHN was relatively concentrated, with lower levels of fluctuation. The box width and 1.5IQR range in 2020 were the largest, and the standard deviation reached 1660.78 pcu, an increase of 41.11% compared with those in 2019, reaching the maximum within the three years. In the first year of the COVID-19 outbreak, affected by the COVID-19 travel restriction policies, the fluctuation of the NPTHN traffic volume reached its maximum. The box width and the 1.5IQR range in 2021 were the narrowest, and the standard deviation was 555.91 pcu, a decrease of 52.77% compared with those in 2019, reaching the minimum within three years. This finding indicates that the impacts of the COVID-19 travel restriction policies on the NPTHN traffic volume were appreciably reduced.

The analysis results in Figure 3 and Table 3 show that the means, medians, and standard deviations of WTIT continued to increase from 2019 to 2021, where the mean in 2020 and 2021 increased by 32.47% and 153.90%, respectively, compared with that of 2019. The corresponding growth rate for the medians was 11.11% and 66.67%, and 75.00% and 467.00% for the standard deviations, respectively. In addition, the box width in 2020 and 2021 increased compared with that in 2019. The frequency of traffic interruption accidents and their fluctuation continued to increase under the background of the COVID-19 travel restriction policies.

The analysis results in Figure 4 and Table 3 show that the mean, medians, and standard deviations of WTCT continued to increase from 2019 to 2021, where the mean in 2020 and 2021 increased by 54.35% and 144.54%, respectively compared with that in 2019. Moreover, the corresponding growth rate for the medians was 77.68% and 142.00%, and 34.00% and 81.00% for the standard deviations, respectively. In addition, the box width and the 1.5IQR range in 2020 and 2021 increased compared with those in 2019 and reached the maximum in 2021. The traffic control time, including its fluctuation, kept increasing year by year in the context of the COVID-19 travel restriction policies.

The analysis results in Table 4 show that in 2020, the proportion of positive growth in the fixed base growth rate of WADT and WTIT was lower than their respective proportion of negative growth in the fixed base growth rate. By contrast, the proportion of positive growth was higher than the proportion of negative growth in the fixed base growth rate in terms of WTCT. In 2021, the proportion of positive growth in the fixed base growth rate of WTCT and WTIT was higher than their respective proportion of negative growth in the fixed base growth rate. Then, the proportion of negative growth was higher than the proportion of positive growth in the fixed base growth rate in terms of WADT.

The analysis of the aforementioned statistical parameters shows that the distribution of the NPTHN traffic volume in Shaanxi Province was relatively stable and the overall fluctuation was small in the year before the COVID-19 outbreak. However, the COVID-19 outbreak brought a series of travel restriction policies, which impacted the levels of traffic volume and caused a large fluctuation of traffic volume, to some extent. Moreover, the pandemic and its travel restriction policies increased the traffic interruption times and traffic control time, including their fluctuation, to a certain extent. The service level of the NPTHN was impacted by the COVID-19 travel restriction policies. The increase in traffic volume fluctuation, traffic interruption times, and traffic control time led to a decrease in the service level of the NPTHN compared with the year before the COVID-19 outbreak. As the pandemic was gradually brought under control, more liberal travel policies were adopted in Shaanxi Province, and the dispersion and fluctuation of the traffic volume decreased. However, those of the traffic interruption times and traffic control time kept increasing, which meant that the service level of the NPTHN did not recover yet.

### 3.2. Analysis of the NPTHN Safety Level

Figure 8, Figure 9 and Figure 10 show the boxplots of the NPTHN safety level assessment parameters (WTA, WTI, and WTD). Figure 11, Figure 12 and Figure 13 depict the dot-line plots of the fixed base growth rate. Table 5 shows the descriptive statistical results of WTA, WTI, and WTD, and Table 6 shows the analysis results of the fixed base growth rate. Table 7 shows the calculation results of the NPTHN accident rate and casualty rate, and Table 8 shows the descriptive statistical results of the NPTHN accident rate and casualty rate.

The analysis results in Figure 8 and Table 5 show that the box width and the 1.5IQR range in 2019 were the narrowest of the three years. This finding indicates that before the COVID-19 outbreak, the distribution of traffic accidents in the NPTHN was relatively concentrated, with smaller fluctuations. The box width in 2020 was the largest, with the fewest outliers. The means, medians, and standard deviations of the WTA kept increasing from 2019 to 2021, where the means in 2020 and 2021 increased by 5.35% and 24.72%, respectively, compared with that of 2019. Moreover, the corresponding growth rate for the medians was 8.33% and 16.67%, and 4.26% and 81.28% for the standard deviations, respectively. This result indicates that the number of traffic accidents and their fluctuation increased continuously in the context of the COVID-19 travel restriction policies.

The analysis results in Figure 9 and Table 5 show that the mean and standard deviations of WTI kept decreasing from 2019 to 2021, whereas the medians remained the same within the three years. Among them, the means in 2020 and 2021 decreased by 2.40% and 18.49% compared with that in 2019, and the corresponding growth rate of the standard deviations was −18.69% and −31.16%, respectively. This result indicates that the number of traffic injuries, including their fluctuation, decreased continuously in the context of the COVID-19 travel restriction policies.

The analysis results in Figure 10 and Table 5 show that the mean and standard deviations of WTD initially increased and then decreased. The growth rate of the mean in 2020 and 2021 was 18.46% and −16.92% compared with that in 2019, whereas the growth rate of the standard deviation was 42.67% and 14.67% compared with that in 2019, respectively. The median of WTD in 2019 was 0.5, which then dropped to 0 in 2020 and 2021, with a decrease of 100%. The number of traffic deaths and its fluctuation increased in the first year of the COVID-19 outbreak and then decreased as the COVID-19 was gradually brought under control.

The analysis results in Table 6 show that the proportion of positive growth in the fixed base growth rate of WTA in 2020 and 2021 was 3.84% and 15.38% higher than its respective proportion of negative growth. By contrast, the proportion of positive growth was 30.77% and 26.92% lower than the proportion of negative growth in the fixed base growth rate in terms of WTD in 2020 and 2021. In 2020, the proportion of positive growth in the fixed base growth rate of WTI was consistent with the proportion of negative growth, both of which were 30.77%. On the contrary, the proportion of negative growth in 2021 was 30.77% higher than that of positive growth in the fixed base growth rate in terms of WTI.

The analysis results in Table 7 and Table 8 show that the annual traffic accident rate of the NPTHN in 2020 and 2021 increased by 17.71% and 55.21%, respectively, compared with that in 2019. Moreover, the mean of the weekly traffic accident rate increased by 16.33% and 52.04%, respectively, compared with that in 2019. After the COVID-19 outbreak, the traffic accident rate of the NPTHN continued to increase within two years under the impact of the COVID-19 travel restriction policies, and the safety level of the NPTHN continued to decrease.

Compared with 2019, the annual traffic injury rate of the NPTHN in 2020 and 2021 increased by 9.09% and 2.27%, the annual traffic death rate increased by 30.00% and 0.00%, and the mean of the weekly traffic injury rate increased by 11.36% and 2.27% respectively. In terms of the weekly traffic injury death rate, the corresponding increases were 30.00% and 0.00%, respectively. After the COVID-19 outbreak, impacted by the COVID-19 travel restriction policies, the traffic accident death rate of the NPTHN appreciably increased in 2020, and the increase of the traffic death rate in 2020 was appreciably higher than that of the traffic injury rate in 2020. The traffic accident injury rate of the NPTHN slightly increased in 2021, and the growth rate decreased compared with 2020. The traffic accident death rate of the NPTHN in 2021 was the same as that in 2019, and the growth rate decreased appreciably compared with that in 2020. Therefore, the traffic accident severity of the NPTHN has been exacerbated appreciably in the first year of the COVID-19 outbreak. With the gradual control of the pandemic, the traffic accident severity decreased to a certain extent.

Compared with 2019, the standard deviation of the weekly traffic accident rate and weekly traffic injury rate in 2020 decreased by 2.63% and 4.00%, respectively, whereas that of the weekly traffic death rate increased by 72.73% compared with that in 2019. This result indicates that the fluctuation of the NPTHN traffic accident rate and the traffic injury rate did not change much after the COVID-19 outbreak, but that of traffic death rate increased appreciably. Compared with 2019, the standard deviation of the weekly traffic accident rate and weekly traffic death rate in 2021 increased by 110.53% and 45.45%, respectively, whereas the standard deviation of the weekly traffic injury rate decreased by 10%. This finding indicates that although COVID-19 was brought under control in 2021, the fluctuation of the traffic accident rate and traffic death rate still increased appreciably, and the fluctuation of the traffic injury rate decreased.

The analysis of the aforementioned statistical parameters shows that the COVID-19 outbreak impacted the safety level of the NPTHN in the Shaanxi Province. The COVID-19 travel restriction policies increased the number of traffic accidents and traffic deaths, including their fluctuation, whereas the number of traffic injuries and its fluctuation decreased continuously in the COVID-19 context. The traffic accident severity of the NPTHN has exacerbated appreciably under the background of the COVID-19 pandemic to some extent. With the gradual control of the pandemic, the traffic accident severity decreased to a certain degree. In summary, the safety level of the NPTHN was impacted by the COVID-19 pandemic and its travel restriction policies. As the pandemic was gradually brought under control, the number of traffic injuries and traffic deaths, including their fluctuation, decreased. However, the number of traffic accidents and their fluctuation remained high. The traffic accident rate kept increasing, whereas the traffic injury rate and the traffic death rate initially increased and then decreased. Both almost recovered to the pre-pandemic level.

### 3.3. Analysis of the NPTHN Operational Orderliness

Table 9 shows the calculation results of the weekly system entropy weight of the NPTHN from 2019 to 2021. Table 10 shows the calculation results of the patency degree, patency degree deviation, and system entropy in each week of the NPTHN from 2019 to 2021. Furthermore, Table 11 shows the descriptive statistical results of system entropy.

The analysis results in Table 10 and Table 11 show that the total entropy of the NPTHN in 2020 was 4.76, with a growth rate of 40.41% compared with that in 2019. Meanwhile, the mean and standard deviation of weekly entropy in 2020 increased by 38.46% and 33.33%, respectively, compared with those in 2019. In the first year of the COVID-19 outbreak, affected by the pandemic and its travel restriction policies, the total entropy of the NPTHN increased appreciably. Moreover, the fluctuation of the NPTHN sharply increased, and the operational orderliness decreased appreciably. The total entropy of the NPTHN in 2021 decreased by 18.91% compared with 2020 but maintained a growth rate of 13.86% compared with that in 2019. With the control of COVID-19, the operational orderliness of the NPTHN improved but has not yet recovered to the level before the pandemic.

In 2021, the average weekly entropy of the NPTHN decreased by 16.67% compared with that in 2020, with a growth rate of 15.38% compared with that in 2019. The standard deviation of the weekly entropy in 2021 decreased by 3.57% compared with that in 2021, with a growth rate of 28.57% compared with that in 2019. With the gradual control of the COVID-19 pandemic, the operational orderliness of the NPTHN improved, its fluctuation diminished, and the overall operation stability of the NPTHN improved. However, a gap still exists from the normal level in 2019.

Based on the above analysis, we have reason to believe that after the COVID-19 outbreak, the operational orderliness of the NPTHN was impacted by the COVID-19 pandemic and its travel restriction policies. Compared with the previous year, the operational orderliness decreased appreciably, its fluctuation increased appreciably, and the overall stability of the NPTHN became poorer. With the gradual control of the pandemic, the operational orderliness of the NPTHN improved in 2021, it has not yet recovered to the normal, pre-pandemic levels.

## 4. Discussion

This study examined the impacts of COVID-19 travel restriction policies on the traffic quality of the NPTHN in the Shaanxi Province. Through the descriptive statistics and dynamic analysis methods, we analyzed the service and safety levels of the NPTHN in the COVID-19 pandemic context. Using dissipative structure theory and entropy theory, based on the perspective of operational orderliness, an assessment model was set up to analyze the operational orderliness of the NPTHN in the COVID-19 pandemic context. Through the above analysis, we confirmed the adverse impacts of the COVID-19 travel restriction policies on the traffic quality of the NPTHN.

The research results show that COVID-19 and its travel restriction policies had an impact on the traffic quality of the NPTHN in Shaanxi Province. The COVID-19 travel restriction policies caused a reduction in the traffic quality of the NPTHN to a certain extent, including the reduction in service level, safety level, and operational orderliness. Among them, the decrease in the NPTHN service level was mainly manifest in the increase in traffic volume fluctuation, traffic interruption times, and traffic control time. The decrease in the NPTHN safety level was mainly reflected in the increase in traffic accidents and traffic deaths, the increase in the traffic accident rate, traffic accident death rate, and the aggravation of traffic accident severity. The decrease in the NPTHN operational orderliness was mainly manifest in the increase in the NPTHN entropy, the decrease in orderliness level, and the intensification of the NPTHN’s operational fluctuation. With the gradual control of the COVID-19 pandemic, the operational orderliness of the NPTHN began to recover, but the service level and safety level did not improve.

This study only examined the limited impacts of the COVID-19 travel restriction policies on the NPTHN traffic quality and could not further explore the deep-seated reasons for these impacts because of the limited data. However, based on the actual situation of the COVID-19 pandemic prevention and control in the Shaanxi Province, referring to the research results of other researchers, we could still undertake some analyses. These analyses could be the reasons for the impacts of the COVID-19 travel restriction policies on the traffic quality of the NPTHN to a certain extent.

First, when COVID-19 broke out, the Shaanxi provincial government adopted strict travel restriction policies, and passenger and freight transportation on the NPTHN could not be carried out normally. With the gradual control of the pandemic, the travel restriction policies were gradually reduced and lifted, resulting in a sharp increase in passenger and freight transportation on the NPTHN. The traffic volume was higher than that of the same period before the pandemic. This event, to some extent, led to a decline in the operational orderliness and stability of the NPTHN, increased the frequency of traffic accidents, and then increased traffic interruptions and traffic control time. As the demand for passenger and freight transportation was alleviated, the traffic volume of the NPTHN gradually reached a level close to that in the previous year, and the fluctuation reduced. Second, the prevention and control measures of COVID-19 brought non-negligible impacts on drivers. For example, long-term home quarantine led to drivers’ depression, which in turn increased dangerous driving behaviors such as speeding and aggressive driving, resulting in an increase in the traffic accident rate. These may be the underlying reasons for the decline of the traffic quality of the NPTHN in the Shaanxi Province in the COVID-19 context.

To summarize, the impacts of the COVID-19 travel restriction policies on the traffic quality of the NPTHN were objective facts. They needed to be identified through data analysis, which was exactly what this study attempted to address. Facing the threat of the COVID-19 pandemic, many countries have not paid enough attention to the impacts of the COVID-19 travel restriction policies on traffic quality. Traffic departments can adopt effective measures to ensure the safe, stable, and orderly operation of the NPTHN in the COVID-19 pandemic context only by fully understanding the impacts of the COVID-19 travel restriction policies on the traffic quality of the NPTHN.

### Limitations

First, considering the limited research materials, we excluded the traffic data for some weeks from 2019 to 2021 to improve the reliability of the research results. We retained the data with overlapping time points in the three years and finally obtained the traffic data of 26 weeks per year as research materials. Although our data processing reduced the sample size, the weekly sequences of these data were completely corresponding, which could ensure the typicality, comparability, and effectiveness of the research materials. Thus, the research results could reflect the impacts of the COVID-19 travel restriction policies on the traffic quality of the NPTHN in Shaanxi Province more truly and effectively. Undoubtedly, the results of this study would be more comprehensive and accurate if the complete traffic data for 52 weeks in the whole year could be used for the study.

Second, the changes of the traffic operation parameters are random to a certain extent, and our research could not rule out the randomness interference completely. However, quantitative analysis and comprehensive comparison of multiple parameters and indicators could reduce the randomness interference to some extent by incorporating the traffic operation data of three consecutive years before and after the COVID-19 outbreak into research.

Finally, this study did not consider factors other than COVID-19 that would impact the traffic quality of the NPTHN. These limitations need to be further considered in future studies to more accurately reflect the traffic quality of the NPTHN in the COVID-19 context.

## 5. Conclusions

Learning the impacts of the COVID-19 travel restriction policies on the traffic quality of the NPTHN is essential. It can help traffic management departments to take effective measures to ensure the traffic quality of the NPTHN in the COVID-19 context, and finally promote the improvement of the service level, safety level, and operational orderliness of the NPTHN. According to the research results of this study, major public health emergencies, such as the COVID-19 pandemic and their traffic restriction policies, will lead to a reduction in the service level, safety level, and operational orderliness of the NPTHN. Therefore, during the pandemic prevention and control period, the traffic management departments should strengthen the control of vehicle operation, such as strictly controlling speeding behavior by strengthening speed restriction measures. The setting strategy of traffic lights on national and provincial trunk highways should be optimized to improve the adaptability and traffic efficiency of road intersections to pedestrians and vehicles based on the actual situation of traffic volume during the pandemic prevention and control period. Optimizing relevant traffic management measures according to the progress of pandemic prevention and control should be considered. These are some effective measures that traffic management departments could take.

The research results of this study are consistent with the actual traffic situation of the NPTHN in Shaanxi Province. The descriptive statistics and dynamic analysis methods can be used to analyze the impacts of the COVID-19 travel restriction policies on the traffic quality of the NPTHN. Introducing the entropy theory and dissipative structure theory to explore the evolution of the NPTHN operational orderliness is effective.

On the one hand, we hope that COVID-19 will be defeated as soon as possible and that similar disasters will not happen again. On the other hand, we also need to be realistic and use more diversified and comprehensive data for preparation. As such, we can deeply explore the impact mechanisms of major public health emergencies, such as the COVID-19 pandemic, on the operation of the road system in future research. Thus, we can provide a theoretical basis for traffic management departments to take effective measures to improve traffic quality.

## Figures and Tables

**Figure 1 ijerph-19-09387-f001:**
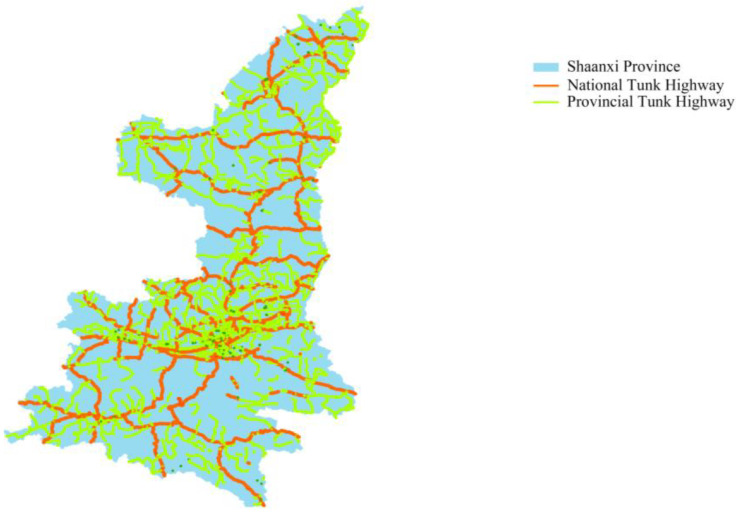
The geographical location overview of the NPTHN of Shaanxi Province.

**Figure 2 ijerph-19-09387-f002:**
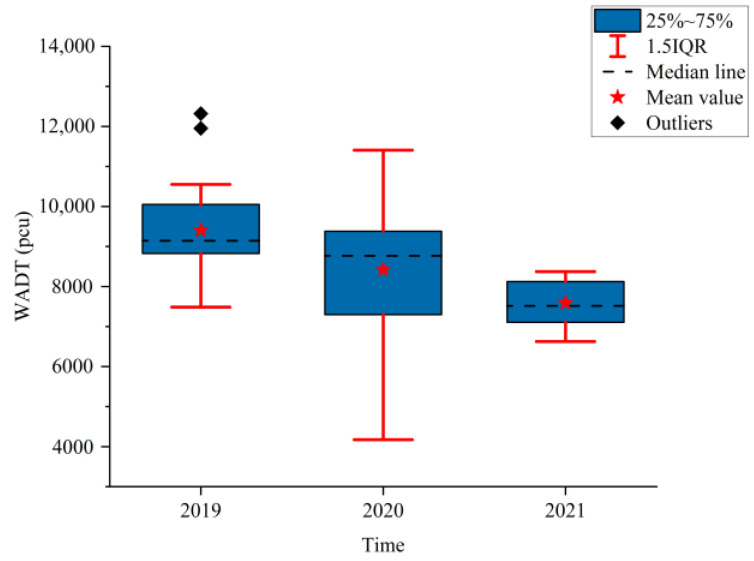
WADT.

**Figure 3 ijerph-19-09387-f003:**
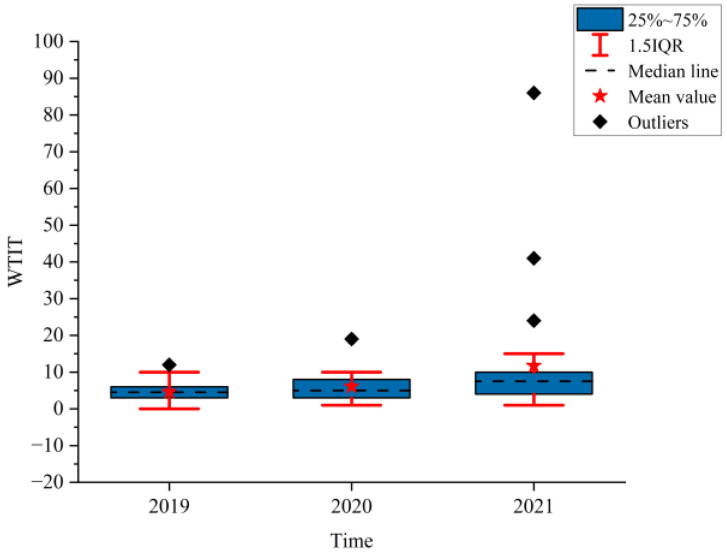
WTIT.

**Figure 4 ijerph-19-09387-f004:**
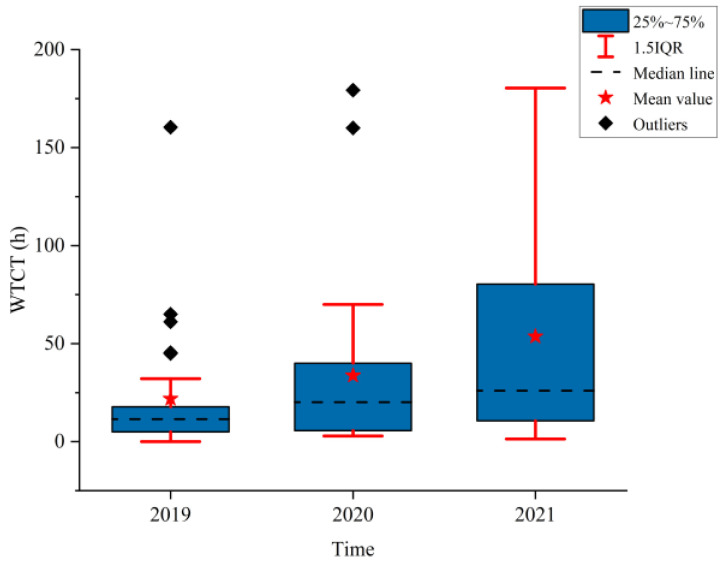
WTCT.

**Figure 5 ijerph-19-09387-f005:**
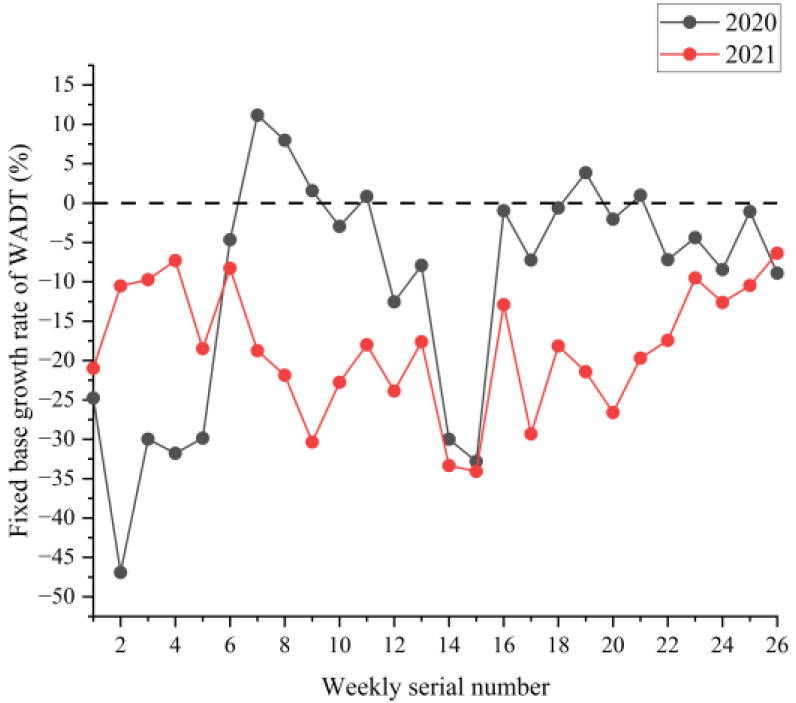
Fixed base growth rate of WADT.

**Figure 6 ijerph-19-09387-f006:**
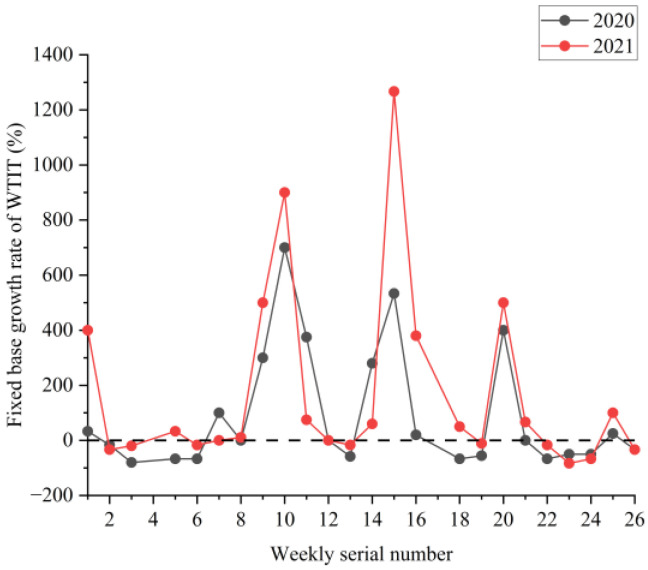
Fixed base growth rate of WTIT.

**Figure 7 ijerph-19-09387-f007:**
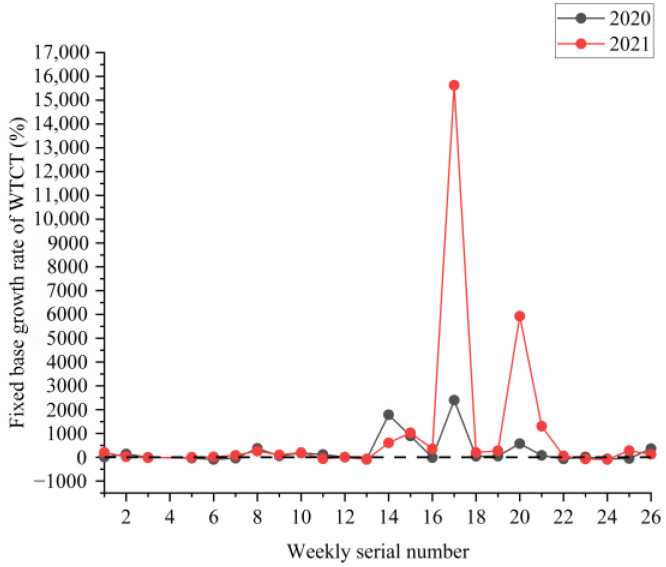
Fixed base growth rate of WTCT.

**Figure 8 ijerph-19-09387-f008:**
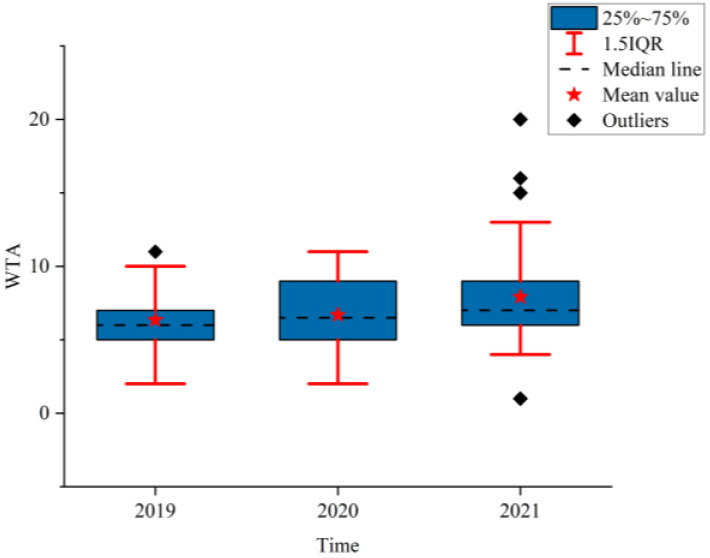
WTA.

**Figure 9 ijerph-19-09387-f009:**
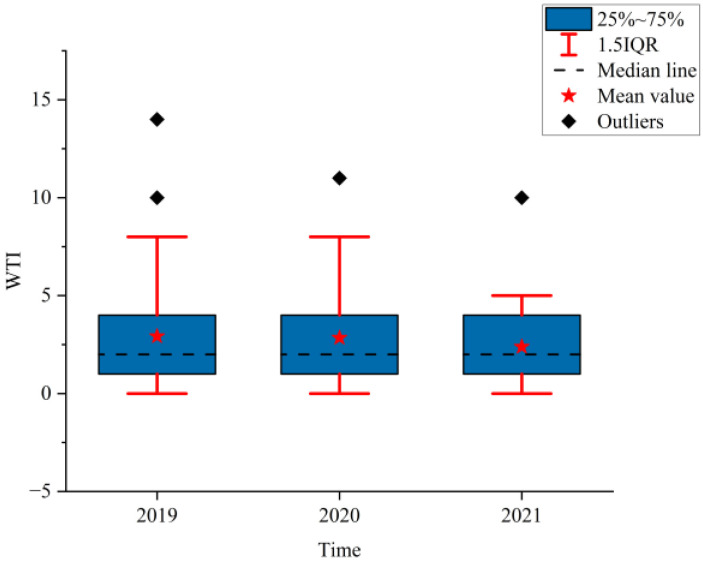
WTI.

**Figure 10 ijerph-19-09387-f010:**
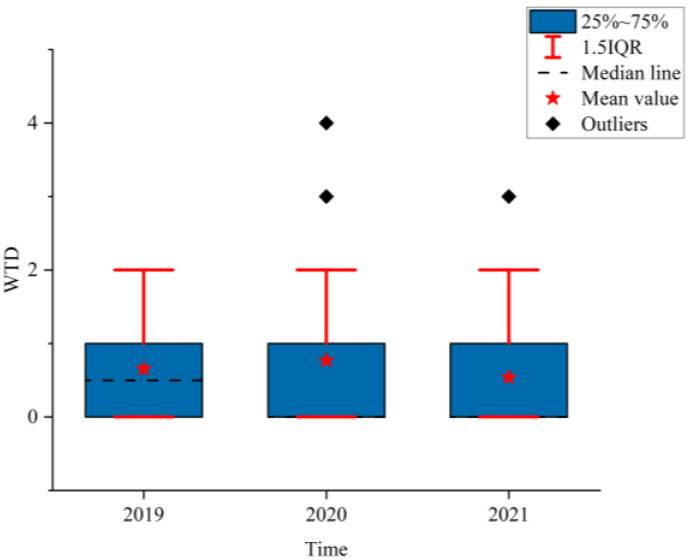
WTD.

**Figure 11 ijerph-19-09387-f011:**
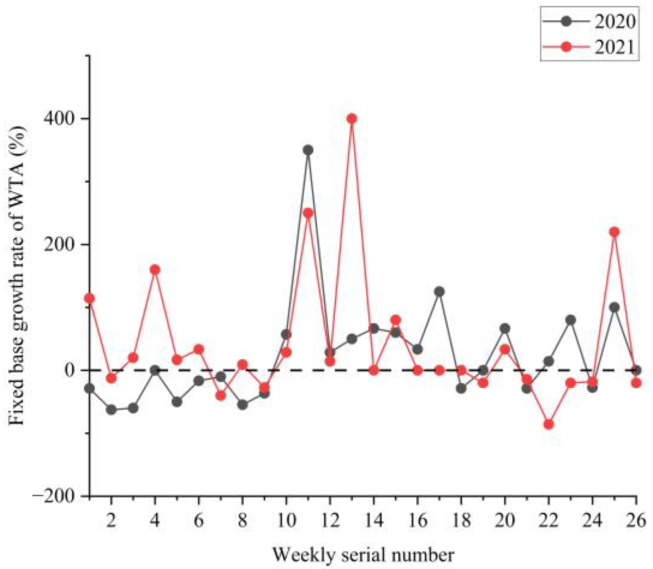
Fixed base growth rate of WTA.

**Figure 12 ijerph-19-09387-f012:**
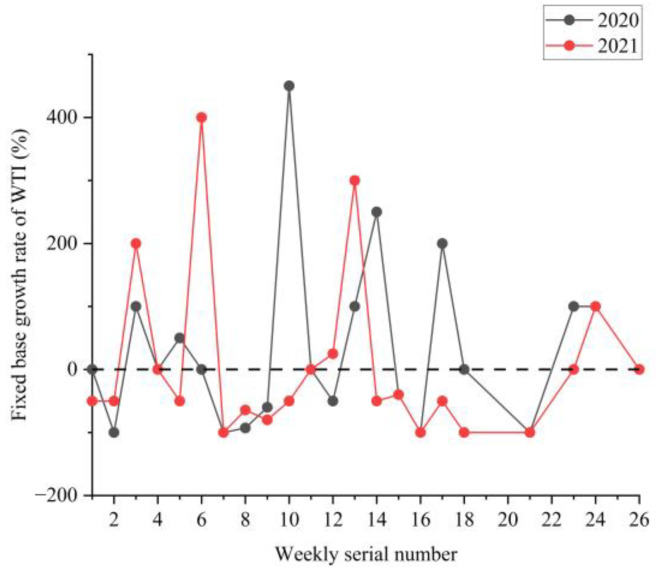
Fixed base growth rate of WTI.

**Figure 13 ijerph-19-09387-f013:**
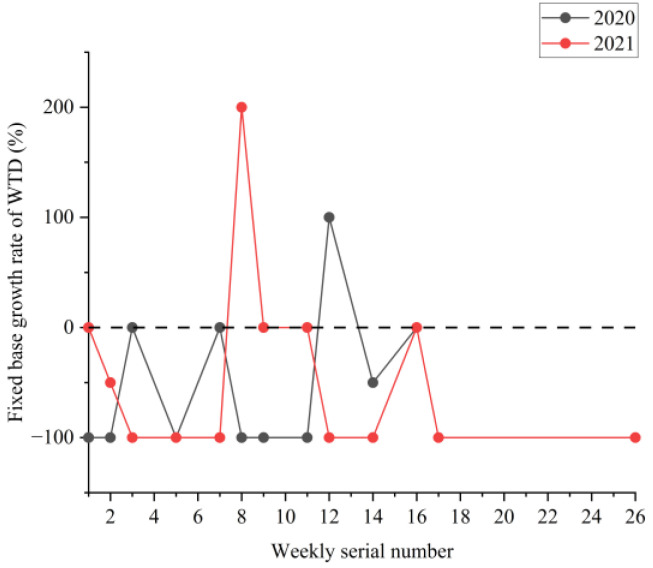
Fixed base growth rate of WTD.

**Table 1 ijerph-19-09387-t001:** Time series analysis dataset of the NPTHN service level assessment parameters.

Weekly Serial Number	WADT (pcu)			WTIT			WTCT (h)		
	2019	2020	2021	2019	2020	2021	2019	2020	2021
1	8384	6307	6625	3	4	15	5.00	5.62	15.00
2	7865	4175	7037	6	5	4	11.95	28.22	14.93
3	8859	6204	7997	5	1	4	3.67	3.50	3.00
4	9032	6161	8372	0	5	7	0.00	15.58	19.37
5	9794	6869	7983	3	1	4	6.00	4.00	6.00
6	9109	8684	8354	6	2	5	45.00	2.87	52.00
7	10,262	11,408	8338	2	4	2	11.50	7.50	21.00
8	9832	10,615	7681	9	9	10	13.22	62.50	47.35
9	10,552	10,718	7349	2	8	12	12.73	20.17	26.00
10	10,368	10,060	8007	1	8	10	7.83	21.83	23.00
11	10,030	10,114	8225	4	19	7	16.60	35.97	6.90
12	10,225	8942	7783	10	10	10	65.00	70.00	65.00
13	9931	9145	8181	12	5	10	160.40	40.00	9.00
14	11,951	8366	7965	5	19	8	9.50	179.33	66.68
15	12,320	8277	8122	3	19	41	16.00	160.00	180.42
16	9473	9381	8250	5	6	24	17.80	15.25	80.38
17	10,051	9323	7104	0	5	86	1.40	35.00	220.20
18	8906	8850	7290	6	2	9	32.12	46.27	97.00
19	9037	9385	7101	9	4	8	45.42	67.62	170.00
20	9168	8980	6729	1	5	6	2.00	13.50	120.53
21	9122	9212	7325	3	3	5	3.17	5.72	44.50
22	8828	8193	7288	6	2	5	61.17	21.50	98.67
23	7972	7623	7214	6	3	1	11.00	12.53	3.33
24	7972	7298	6966	6	3	2	14.63	4.08	1.33
25	7995	7908	7158	4	5	8	8.12	4.00	30.75
26	7484	6817	7007	3	2	2	4.75	22.00	10.67

**Table 2 ijerph-19-09387-t002:** Time series analysis dataset of the NPTHN safety level assessment parameters.

Weekly Serial Number	WTA			WTI			WTD		
	2019	2020	2021	2019	2020	2021	2019	2020	2021
1	7	5	15	8	8	4	1	0	1
2	8	3	7	6	0	3	2	0	1
3	5	2	6	1	2	3	1	1	0
4	5	5	13	4	4	4	0	0	1
5	6	3	7	2	3	1	1	0	0
6	6	5	8	1	1	5	0	0	0
7	10	9	6	2	0	0	1	1	0
8	11	5	12	14	1	5	1	0	3
9	11	7	8	10	4	2	1	0	1
10	7	11	9	2	11	1	0	1	0
11	2	9	7	3	3	3	2	0	2
12	7	9	8	4	2	5	1	2	0
13	4	6	20	1	2	4	0	0	2
14	6	10	6	2	7	1	2	1	0
15	5	8	9	5	3	3	0	2	0
16	6	8	6	4	0	0	1	1	1
17	4	9	4	2	6	1	2	0	0
18	7	5	7	1	1	0	0	1	0
19	5	5	4	0	4	0	0	2	0
20	3	5	4	0	2	0	0	0	0
21	7	5	6	1	0	0	0	3	0
22	7	8	1	0	0	3	0	1	0
23	5	9	4	1	2	1	0	4	0
24	11	8	9	1	2	2	0	0	2
25	5	10	16	0	5	10	0	0	0
26	5	5	4	1	1	1	1	0	0

**Table 3 ijerph-19-09387-t003:** The descriptive statistical results of the NPTHN service level assessment parameters.

Parameter	Mean Value			Standard Deviation		
	2019	2020	2021	2019	2020	2021
WADT (pcu)	9404.69	8423.65	7594.27	1176.94	1660.78	555.91
WTIT	4.62	6.12	11.73	3.03	5.29	17.19
WTCT (h)	22.54	34.79	55.12	33.31	44.52	60.13

**Table 4 ijerph-19-09387-t004:** The analysis results of the fixed base growth rate of the NPTHN service level assessment parameters.

Parameter	Time	Positive Growth Times	Proportion (%)	Negative Growth Times	Proportion (%)	Zero Growth Times	Proportion (%)
WADT	2020	6	23.08	20	76.92	0	0.00
	2021	0	0.00	26	100.00	0	0.00
WTIT	2020	10	38.46	11	42.31	3	11.54
	2021	13	50.00	9	34.62	2	7.69
WTCT	2020	16	61.54	9	34.62	0	0.00
	2021	18	69.23	5	19.23	2	7.69

Note: We excluded cases for which the relevant parameter value in 2019 was 0.

**Table 5 ijerph-19-09387-t005:** The descriptive statistical results of the NPTHN safety level assessment parameters.

Parameter	Mean Value			Standard Deviation		
	2019	2020	2021	2019	2020	2021
WTA	6.35	6.69	7.92	2.35	2.45	4.26
WTI	2.92	2.85	2.38	3.37	2.74	2.32
WTD	0.65	0.77	0.54	0.75	1.07	0.86

**Table 6 ijerph-19-09387-t006:** The analysis results of the fixed base growth rate of the NPTHN safety level assessment parameters.

Parameter	Time	Positive Growth Times	Proportion (%)	Negative Growth Times	Proportion (%)	Zero Growth Times	Proportion (%)
WTA	2020	12	46.15	11	42.31	3	11.54
	2021	13	50.00	9	34.62	4	15.38
WTI	2020	8	30.77	8	30.77	6	23.08
	2021	5	19.23	13	50.00	4	15.39
WTD	2020	1	3.85	9	34.62	3	11.54
	2021	1	3.85	8	30.77	4	15.39

Note: We excluded cases for which the relevant parameter value in 2019 was 0.

**Table 7 ijerph-19-09387-t007:** The calculation results of the NPTHN accident rate and casualty rate.

Weekly Serial Number	Weekly Traffic Accident Rate (1/10,000)			Weekly Traffic Injury Rate (1/10,000)			Weekly Traffic Death Rate (1/10,000)		
	2019	2020	2021	2019	2020	2021	2019	2020	2021
1	1.19	1.13	3.23	1.36	1.81	0.86	0.17	0.00	0.22
2	1.45	1.03	1.42	1.09	0.00	0.61	0.36	0.00	0.20
3	0.81	0.46	1.07	0.16	0.46	0.54	0.16	0.23	0.00
4	0.79	1.16	2.22	0.63	0.93	0.68	0.00	0.00	0.17
5	0.88	0.62	1.25	0.29	0.62	0.18	0.15	0.00	0.00
6	0.94	0.82	1.37	0.16	0.16	0.86	0.00	0.00	0.00
7	1.39	1.13	1.03	0.28	0.00	0.00	0.14	0.13	0.00
8	1.60	0.67	2.23	2.03	0.13	0.93	0.15	0.00	0.56
9	1.49	0.93	1.56	1.35	0.53	0.39	0.14	0.00	0.19
10	0.96	1.56	1.61	0.28	1.56	0.18	0.00	0.14	0.00
11	0.28	1.27	1.22	0.43	0.42	0.52	0.28	0.00	0.35
12	0.98	1.44	1.47	0.56	0.32	0.92	0.14	0.32	0.00
13	0.58	0.94	3.49	0.14	0.31	0.70	0.00	0.00	0.35
14	0.72	1.71	1.08	0.24	1.20	0.18	0.24	0.17	0.00
15	0.58	1.38	1.58	0.58	0.52	0.53	0.00	0.35	0.00
16	0.90	1.22	1.04	0.60	0.00	0.00	0.15	0.15	0.17
17	0.57	1.38	0.80	0.28	0.92	0.20	0.28	0.00	0.00
18	1.12	0.81	1.37	0.16	0.16	0.00	0.00	0.16	0.00
19	0.79	0.76	0.80	0.00	0.61	0.00	0.00	0.30	0.00
20	0.47	0.80	0.85	0.00	0.32	0.00	0.00	0.00	0.00
21	1.10	0.78	1.17	0.16	0.00	0.00	0.00	0.47	0.00
22	1.13	1.39	0.20	0.00	0.00	0.59	0.00	0.17	0.00
23	0.90	1.69	0.79	0.18	0.37	0.20	0.00	0.75	0.00
24	1.97	1.57	1.85	0.18	0.39	0.41	0.00	0.00	0.41
25	0.89	1.81	3.19	0.00	0.90	2.00	0.00	0.00	0.00
26	0.95	1.05	0.82	0.19	0.21	0.20	0.19	0.00	0.00
Total for the year	0.96	1.13	1.49	0.44	0.48	0.45	0.10	0.13	0.10

**Table 8 ijerph-19-09387-t008:** The descriptive statistical results of the NPTHN accident rate and casualty rate.

Index	Mean Value (1/10,000)			Standard Deviation (1/10,000)		
	2019	2020	2021	2019	2020	2021
Weekly traffic accident rate	0.98	1.14	1.49	0.38	0.37	0.80
Weekly traffic injury rate	0.44	0.49	0.45	0.50	0.48	0.45
Weekly traffic death rate	0.10	0.13	0.10	0.11	0.19	0.16

**Table 9 ijerph-19-09387-t009:** Calculation results of the weekly system entropy weight of the NPTHN.

Weekly Serial Number	K_i_			Weekly Serial Number	K_i_		
	2019	2020	2021		2019	2020	2021
1	0.03	0.03	0.05	14	0.04	0.12	0.03
2	0.05	0.03	0.01	15	0.03	0.12	0.13
3	0.04	0.01	0.01	16	0.04	0.04	0.08
4	0.00	0.03	0.02	17	0.00	0.03	0.28
5	0.03	0.01	0.01	18	0.05	0.01	0.03
6	0.05	0.01	0.02	19	0.08	0.03	0.03
7	0.02	0.03	0.01	20	0.01	0.03	0.02
8	0.08	0.06	0.03	21	0.03	0.02	0.02
9	0.02	0.05	0.04	22	0.05	0.01	0.02
10	0.01	0.05	0.03	23	0.05	0.02	0.00
11	0.03	0.12	0.02	24	0.05	0.02	0.01
12	0.08	0.06	0.03	25	0.03	0.03	0.03
13	0.10	0.03	0.03	26	0.03	0.01	0.01

**Table 10 ijerph-19-09387-t010:** Calculation results of the patency degree, patency degree deviation, and system entropy in each week of the NPTHN from 2019 to 2021.

Weekly Serial Number	*λ_i_*			Δ*λ**_i_*			*S_i_*(*t*)		
	2019	2020	2021	2019	2020	2021	2019	2020	2021
1	0.89	0.75	0.87	0.00	0.00	0.00	0.00	0.00	0.00
2	0.84	0.50	0.93	0.00	0.00	0.00	0.00	0.00	0.00
3	0.94	0.74	1.05	0.00	0.00	0.05	0.00	0.00	0.10
4	0.96	0.73	1.10	0.00	0.00	0.10	0.00	0.00	0.25
5	1.04	0.82	1.05	0.04	0.00	0.05	0.12	0.00	0.10
6	0.97	1.03	1.10	0.00	0.03	0.10	0.00	0.04	0.18
7	1.09	1.35	1.10	0.09	0.35	0.10	0.14	0.33	0.07
8	1.05	1.26	1.01	0.05	0.26	0.01	0.39	0.65	0.08
9	1.12	1.27	0.97	0.12	0.27	0.00	0.16	0.59	0.00
10	1.10	1.19	1.05	0.10	0.19	0.05	0.07	0.49	0.25
11	1.07	1.20	1.08	0.07	0.20	0.08	0.22	1.20	0.22
12	1.09	1.06	1.02	0.09	0.06	0.02	0.67	0.29	0.15
13	1.06	1.09	1.08	0.06	0.09	0.08	0.60	0.18	0.31
14	1.27	0.99	1.05	0.27	0.00	0.05	0.60	0.00	0.19
15	1.31	0.98	1.07	0.31	0.00	0.07	0.37	0.00	1.19
16	1.01	1.11	1.09	0.01	0.11	0.09	0.05	0.27	0.79
17	1.07	1.11	0.94	0.07	0.11	0.00	0.00	0.21	0.00
18	0.95	1.05	0.96	0.00	0.05	0.00	0.00	0.05	0.00
19	0.96	1.11	0.94	0.00	0.11	0.00	0.00	0.18	0.00
20	0.97	1.07	0.89	0.00	0.07	0.00	0.00	0.15	0.00
21	0.97	1.09	0.96	0.00	0.09	0.00	0.00	0.12	0.00
22	0.94	0.97	0.96	0.00	0.00	0.00	0.00	0.00	0.00
23	0.85	0.90	0.95	0.00	0.00	0.00	0.00	0.00	0.00
24	0.85	0.87	0.92	0.00	0.00	0.00	0.00	0.00	0.00
25	0.85	0.94	0.94	0.00	0.00	0.00	0.00	0.00	0.00
26	0.80	0.81	0.92	0.00	0.00	0.00	0.00	0.00	0.00
∑							3.39	4.76	3.86

**Table 11 ijerph-19-09387-t011:** The descriptive statistical results of system entropy.

Time	Minimum Value	Maximum Value	Mean Value	Standard Deviation
2019	0.00	0.67	0.13	0.21
2020	0.00	1.20	0.18	0.28
2021	0.00	1.19	0.15	0.27

## Data Availability

Publicly available datasets were analyzed in this study. This data can be found here: https://jtyst.shaanxi.gov.cn/news/570526375327/3.htm (accessed on 10 June 2022).

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
