# Peer review of "Impacts of COVID-19 Travel Restriction Policies on the Traffic Quality of the National and Provincial Trunk Highway Network: A Case Study of Shaanxi Province"

_ijerph, 2022, doi:10.3390/ijerph19159387_

Round 1

Reviewer 1 Report

The authors have presented a very study about the impact of  COVID-19 travel restriction policies on traffic quality of the national and provincial trunk highway network.

I recommend the authors to enrich the introduction by including more recent literatures/

Author Response

Yours sincerely.

Reviewer 2 Report

The manuscript deals with an interesting topic so it has potential for publication since it is related to the impacts of the COVID-19 pandemic. However, its current state has several points that need to be improved before it can be accepted for publication.

The first point is the research contribution since it focuses on identifying the impacts on traffic quality caused by restrictive transport policies implemented after the COVID break out. The problem is that in the same introduction, the authors establish that several investigations have already been carried out and it is possible to affirm that the COVID pandemic had “non-negligible” impacts on traffic quality. Therefore, in the same introduction, they are responding to the objective of the investigation. This does not mean that the article does not make sense, however, the objective must be reconsidered so that it is more specific and allows a real contribution to be visualized.

The article is a case study (Shaanxi Province and its NPTHN) however, it is not described correctly. It is necessary to include a section describing the case study that allows the reader to size and understands the context of the study area. It is suggested to show the case study on a map.

Likewise, the manuscript focuses on studying and discussing the impacts of COVID travel restriction policies applied in the case study, however, these policies are only mentioned in general terms. Therefore, it is necessary to describe these restriction policies in greater detail to obtain better conclusions.

By describing in detail the case study and the restrictive policies implemented, perhaps the objective of the research and its contribution could be improved (e.g. identify the impacts of certain types of traffic restriction policies, in regions/provinces with similar characteristics to those of the case study). At this time it is not possible to do this, since it is not clear what the characteristics of the province are, nor of its trunk network, or of the implemented policies.

On the other hand, the little literature review presented is scarce (mostly related to impacts of COVID on traffic). It is necessary to include a more extensive literature review that shows why this study is relevant (perhaps a gap in the literature can be identified that this research can address) and justifies the proposed method, especially the fact of applying the entropy theory to analyze a phenomenon as complex as traffic.

Also, it is necessary to describe the observation sites with some level of detail (characteristics, equipment, location, etc.).

Regarding the parameters analyzed, it is necessary to define for this investigation what is meant by "Weekly Traffic Interruption Times (WTIT)" and "Weekly Traffic Control Time (WTCT)" (there is not even mention of the units for WTIT). Likewise, it must also be clearly defined what is meant by "operation orderliness” (the definition is mentioned up to line 140 in section 2.2.2, but it would be better to be presented before).

In various parts of the manuscript, the same ideas are repeated, particularly in sections 3, 4, and 5. In section 3 (results) a brief discussion of findings is presented and later, section 4 (discussion) raises the same ideas in a little more detail. Likewise, section 5 (conclusions) is very similar to section 4. Basically, these sections establish that the COVID-19 restrictive policies implemented in Shaanxi Province had negative impacts on traffic quality when analyzing the changes from 2019 to 2020; and for 2021, with the removal of restrictive policies, the traffic quality situation has improved but has not yet returned to its pre-pandemic state.

In this sense, conclusions must address something else, such as how this research's findings can help improve decision-making. Regarding this, line 485 mentions that “… helps the traffic management departments to take effective measures to ensure traffic quality…”. However, it is not clear how the findings, as are presented at this moment, could help to do so. This should be addressed in greater detail in the conclusion.

Finally, the article requires considerable improvements in English writing, since it has some deficiencies that make reading difficult and dense. These are some examples:

- Lines 36 – 39: It is not necessary to repeat NPTHN in the same idea.

- Lines 39-45: The order of ideas is confusing since previous studies are mentioned that establish impacts of the pandemic on traffic, but first it speaks of impacts on the severity of accidents, then it establishes that it affected traffic volume, and later again mentions aspects of accidents and safety.

- Line 49: It is not necessary to say “Using the methods of statistical analysis and modeling…”, it is easier to read “using statistical analysis and modeling".

- Line 113: “dissipative structure” is included 3 times in a single 3-line paragraph.

- Lines 130 -132: “discrete information source” is repeated 3 times in a 2-line paragraph.

- Lines 140-161: It is just one paragraph with more than 20 lines, this makes it difficult for the reader to catch the idea

- Lines 335-360: It is a single paragraph of 25 lines. This is too long and difficult to read, not just for the extension, but also for its repetitive wording (a lot of percentages, several "compared with 2019...").

- Lines 343 and 343: There is just a one-word difference in two consecutive lines, this exemplifies the repetitive wording.

These are just some examples of writing improvement opportunities, however, the entire manuscript needs to be reviewed to eliminate repetitive wording and accomplish an easy-of-read article.

Author Response

Yours sincerely

Reviewer 3 Report

The paper presents an analysis of Covid-19 travel restrictions on road traffic for the national and provincial trunk highway network (NPTHN) in Shaanxi Province, based on aggregated traffic data collected in that region over the course of three years, comparing the last year before the pandemic (2019) as a baseline to the following two years. The authors introduce multiple KPIs to draw conclusions in terms of service level, safety, and operation orderliness. While the former two are mainly based on descriptive statistic analysis, the latter is based on an introduced assessment model derived from entropy theory.

The article is well organized and contains elements one expects from scientific work. The sections are well developed and presented in a very understandable manner. They present the base data, evaluations, interpretations, and conclusions.

Overall a very solid paper certainly worth publishing however, allow me to make two remarks, one major and one minor:

Major remark:

I found an error in your evaluation/assessment. In line 344f, you write:

“[…] and the mean of the weekly traffic death rate 344

increased by 72.73% and 45.45% respectively.”

You base this on Table 8. However, you mixed up the numbers from the mean values  with the numbers from the standard deviation. If you consider the numbers from the mean values (0.1; 0.13; 0.1), you get an increase of 18.18% and 0%, respectively, which would make the following conclusion in lines questionable, and you might want to rephrase 345-348:

“It can be seen that after the COVID-19 out- 345
break, impacted by the COVID-19 travel restriction policies, the traffic accident death rate 346
of the NPTHN significantly increased, and the increase of the traffic death rate was sig- 347
nificantly higher than that of the traffic injury rate.”

This actually brings me to my minor remark:

You use the term “significantly” quite often in your assessments. As you use means of statistics in your paper, I am sure that you are aware, that this term invokes a certain expectation in the reader, as it is a well-known term in that area (compare:  https://en.wikipedia.org/wiki/Statistical_significance )

To back your claims of “significance” in the data analysis, I suggest you incorporate the respective tests and give the alpha and p-values you used to back those assessments. Otherwise, you might want to rephrase and use different terms.

Author Response

Yours sincerely

Round 2

Reviewer 2 Report

The manuscript has been improved by addressing the points previously mentioned and I recommend it is accepted for publication.

English writing is acceptable and just minor spell improvements may be necessary.

Lines 103 to 108 can be optimized in order to avoid repeating "...analyze the traffic quality of the NPTHN" (this is mentioned in line 105 and lines 107-108).